# Cancers after Chornobyl: From Epidemiology to Molecular Quantification

**DOI:** 10.3390/cancers11091291

**Published:** 2019-09-02

**Authors:** Dimitry Bazyka, Natalya Gudzenko, Iryna Dyagil, Iryna Ilienko, David Belyi, Vadim Chumak, Anatoly Prysyazhnyuk, Elena Bakhanova

**Affiliations:** National Research Center for Radiation Medicine, 53 Melnikov str, Kyiv 04050, Ukraine

**Keywords:** Chornobyl (Chernobyl), cleanup workers, leukemia, thyroid cancer, breast cancer, telomere length

## Abstract

An overview and new data are presented from cancer studies of the most exposed groups of the population after the Chornobyl accident, performed at the National Research Center for Radiation Medicine (NRCRM). Incidence rates of solid cancers were analyzed for the 1990–2016 period in cleanup workers, evacuees, and the general population from the contaminated areas. In male cleanup workers, the significant increase in rates was demonstrated for cancers in total, leukemia, lymphoma, and thyroid cancer, as well as breast cancer rates were increased in females. Significantly elevated thyroid cancer incidence was identified in the male cleanup workers cohort (150,813) in 1986–2012 with an overall standardized incidence ratio (SIR) of 3.35 (95% CI: 2.91–3.80). A slight decrease in incidence rates was registered starting at 25 years after exposure. In total, 32 of 57 deaths in a group of cleanup workers with confirmed acute radiation syndrome (ARS) or not confirmed ARS (ARS NC) were due to blood malignancies or cancer. Molecular studies in cohort members included gene expression and polymorphism, FISH, relative telomere length, immunophenotype, micronuclei test, histone H2AX, and TORCH infections. Analysis of chronic lymphocytic leukemia (CLL) cases from the cohort showed more frequent mutations in telomere maintenance pathway genes as compared with unexposed CLL patients.

## 1. Introduction

Exposure to ionizing radiation is associated with increased risk of cancer—primarily leukemia, thyroid, and breast cancer. The first reports on the effects of radiation exposure were published for the Japanese A-bomb survivors [1,2].

Later analysis showed an increased solid cancer incidence among the Life Span Study (LSS) atomic bomb survivors in Hiroshima and Nagasaki, using the updated case numbers and dosimetry in a cohort of 105,444 subjects. For females, the dose response was consistent with linearity, with an estimated excess relative risk (ERR) of 0.64 per Gy (95% CI: 0.52 to 0.77); for males, an ERR of 0.20 (95% CI: 0.12 to 0.28) at 1 Gy was demonstrated [3].

The leukemia results indicated that there was a nonlinear dose response for leukemias other than chronic lymphocytic leukemia or adult T-cell leukemia, which varied markedly with time and age at exposure. Although the leukemia excess risks generally declined with attained age or time since exposure, there was evidence that the radiation-associated excess leukemia risks, especially for acute myeloid leukemia, had persisted throughout the follow-up period out to 55 years after the bombings [4]. Chronic lymphocytic leukemia (CLL) risks were not analyzed as such pathology is absent in Japan and it has not occurred among radiation-exposed.

These effects were then confirmed by the numerous studies of populations exposed to medical or occupational radiation [5,6,7]. A study of associations between ionizing radiation and site-specific solid cancer mortality was performed among 308,297 nuclear workers employed in France, the United Kingdom, and the United States. The risks were shown for non-CLL leukaemia with an ERR of 2.96 per Gy, (90% CI = 1.17; 5.21), and between cumulative dose and mortality from solid cancers, with an ERR of 0.47 per Gy, (90% CI = 0.18; 0.79) Using a maximum-likelihood method, an attempt was made with the same cohort to quantify associations between radiation dose- and site-specific cancer [8,9]. Positive point estimates were obtained for lung, colon, and prostate cancers. Most of these estimated coefficients exhibited substantial imprecision.

Further evidence is needed regarding associations between cancer and low-dose radiation, and Chornobyl data could provide it.

Chornobyl findings were analyzed in the 2008 report of United Nations Scientific Committee on the Effects of Atomic Radiation (UNSCEAR) [10]. Among cancers, only an increase of thyroid cancer rates was confirmed in those exposed at childhood. However, results of only a few cancer studies were available and analyzed at that time.

Studies of the general population exposed to 131I after the Chornobyl nuclear accident have demonstrated significant excess of thyroid cancers after exposure in childhood, which was substantially greater than originally expected [11]. Data on those exposed in adulthood are more controversial. Chornobyl cleanup workers likely experienced an increased risk of leukemia, already known to be a radiogenic cancer. Other possible health effects, which were not expected, include a possible increased risk for chronic lymphocytic leukemia (CLL), non-Hodgkin lymphoma (NHL), multiple myeloma, and thyroid cancer after adulthood exposure [11]. There are indications of the excess of radiogenic breast cancer that need further research. However, it is still not possible to separate radiogenic cancers from spontaneous cancers using the specific markers and epidemiology data could be questioned, as that led to the opinion of overestimation of Chornobyl effects on health [12]. The aim of the paper is to analyze the results of the national and international studies conducted at the National Research Center for Radiation Medicine (NRCRM) on health effects following the Chornobyl accident in Ukraine.

## 2. Study Subjects and Methods

A retrospective cancer incidence study was performed in cohorts of cleanup workers, evacuees from the 30 km exclusion zone, and different groups of the exposed population using the data of the Chornobyl State Registry of Ukraine (SRU). Their health status is being monitored in the local hospitals according to the national follow-up program [13]. As of 30 June 2015, a total of 318,988 cleanup workers were registered in the registry. In total, 196,423 of them were males involved in cleanup in 1986–1987 and 11,300 were females.

The data from the SRU were used to investigate the cancer incidence among the Chornobyl accident cleanup workers of 1986–1987 and among evacuees. The personified data of SRU on cancer patients were compared with the database of the National Cancer Registry of Ukraine (NCRU), providing the possibility to exclude all cases with incomplete verification of diagnosis and doubled data. After this procedure, all duplicates and cases without validated diagnosis were eliminated.

Notification of cancer cases has been mandatory in Ukraine since 1953, with the NCRU established in 1996. Cancer registration procedures at the NCRU are in accordance with international standards and recommendations. The proportion of microscopically verified cases increased from 73.6% in 2002 to 82.3% in 2012, with death-certificate-only proportions stable at around 0.1% and unknown stage recorded in 9.6% of male and 7.5% of female solid tumors. Timeliness was considered acceptable, with reporting >99% complete within a turn-around time of 15 months [14].

Linkage of SRU data with the NCRUdatabase made it possible to monitor incidence of malignant tumors of different sites in cleanup worker cohorts with an acceptable degree of confidence, and to formulate the hypotheses to test them in analytical studies.

At the territories contaminated with radionuclides after Chornobyl, collection of information on all cancer cases was performed in the Luginy, Narodichy, and Ovruch districts of the Zhitomir region, and the Borodyanka, Ivankov, and Polesskoye districts of the Kyiv region. For the data collection all relevant medical documents (including emergency notifications of new cancer cases as well as death certificates) were obtained from all medical institutions where these patients were diagnosed and treated.

Age specific and age standardized incidence rates were calculated for the period 1990–2016 and compared with respective data on Ukraine as a whole and also on the Zhitomir and Kyiv oblasts as those include the most contaminated territories. Age distribution of the world standard population was used as a standard.

One of the priority groups under the NRCRM follow-up included 190 from 237 cleanup workers who were citizens of Ukraine diagnosed with acute radiation syndrome (ARS). This group consisted of 91 confirmed ARS survivors and 99 patients who had no all symptoms of ARS according to current classification (ARS non-confirmed—NC) [15]. Doses of ARS patients, which were determined by biodosimetry methods after the accident, ranged from 0.1 to 7.1 Gy in ARS survivors and 0.1–1.0 Gy in ARS NC group [16].

In parallel to epidemiological studies a set of molecular investigations was performed among the Cohorts members. A study was performed on 235 male Chornobyl accident cleanup workers exposed in 1986–1987 (doses of external exposure: (M ± SD: 419.48 ± 654.60; range 0.10–3500 mSv)); mean age 58.34 ± 6.57 years. A control group included 45 nonexposed subjects (mean age: 50.60 ± 5.37 (M ± SD)). Gene expression was performed by RT-PCR on 7900 HT Analyzer using TLDA for *BCL2*, *CDKN2A*, *CLSTN2*, *GSTM1*, *IFNG*, *IL1B*, *MCF2L*, *SERPINB9*, *STAT3*, *TERF1*, *TERF2*, *TERT*, *TNF*, *TP53*, and *CCND1* genes. Relative telomere length (RTL) was analyzed by flow-FISH; immune cell subset expression, γ-H2AX; and CyclinD1, was measured by flow cytometry.

## 3. Results

Epidemiological studies on the health effects of the Chornobyl accident were initiated in collaboration with the international scientific community and were concentrated firstly on leukemia and thyroid cancer risks, accounting for their widely recognized link to ionizing radiation exposure.

Cancer effects in the general population were studied in inhabitants of radioactively contaminated areas of the Zhytomyr and Kyiv regions of Ukraine. In 1986, the cohort included 360,700 people exposed to radiation. In 2016, data were available on 170,600 individuals. The mean effective dose of external exposure was 22.4 mSv and thyroid dose was from 187 to 221 mGy. The number of cancers registered in the National Cancer Registry of Ukraine included 26,979 cases. We have not revealed an excess of solid cancers in total for all the period. Increased rates were registered for thyroid cancer. The expected number of thyroid cancer cases in a cohort for 1990–2016 period was 347.5, the observed number was 450 cases (SIR 129.5; 95% CI 117.5–141.5).

A cohort of evacuees from Prypiat, Chornobyl towns, and the 30 km exclusion zone were studied for the 1990–2016 period, which included 50,700 subjects in 1990 and 67,200 subjects in 2016, mainly due to continuation of relocation from areas adjacent to the 30 km zone. Doses of external exposure varied from 10 to 30 mSv, thyroid doses were in the range of 184.4–857.5 mGy. The number of registered solid cancer cases was 4,116, which was lower than expected according to the national standard. The number of observed thyroid cancer cases was 346 compared with 85.7 expected (SIR 403.7; 95% CI 361.2–446.3). Workers from Ukraine who participated in emergency response and cleanup were exposed to the highest doses [17]. As a result, they were expected to be the most affected population group in terms of radiation-induced cancers, and possibly noncancer diseases including cardiovascular ones.

Epidemiology studies in cleanup workers of 1986–1987 have shown an increased incidence of some cancer types diagnosed in 1994–2016. The follow-up continues. Among nosology forms, the most notable are leukemia, thyroid cancer (both genders), and breast cancer in female cleanup workers. A total cancer incidence exceeded national levels during the postaccident period up to 2005. Starting from the 2006 the incidence rates differ from the national standards insignificantly (Table 1).

### Cancers and Leukemia in ARS Survivors

The NRCRM follow-up of those most exposed included a total of 190 subjects diagnosed with the acute radiation syndrome in 1986. From them, 91 diagnoses were confirmed during the re-evaluation three years later (ARS 1–3) and 99 were not confirmed (ARS NC). During the follow-up 57 patients had died in total, including 32 deaths that were caused by solid cancers or blood malignancies.

The first case of solid cancer was diagnosed 6 years after exposure in an ARS NC survivor. It is necessary to note that all following cases of solid tumors either developed without any clinical symptoms and were revealed by chance during routine examination or, following minimum nonspecific complaints (so called “syndrome of minor signs”), were revealed due to physicians’ oncological experience. Over 33 years of follow-up, solid cancers developed in 12 ARS1-3 survivors and 12 patients of ARS-NC group. In the first group three cases of basal cell carcinoma (D04.4, D04.7), two prostate cancers (C61) (in one patient prostate cancer had combined with basal cell carcinoma), two thyroid cancers (C73), one case of urinary bladder (C67.8), one case of colon cancer (C18.7), one case of liver cancer (C22.0), one case of maxillary sinus cancer (C31.0), one tumor of the right cerebellopontine angle (C71.6), and one mandible neuroma with malignant transformation (C72.5) were detected. Four patients of ARS-NC were diagnosed with gastric cancer (C16.9), three colon cancers (C18.9), one case of kidney (C64), one case of prostate (C61), one case of lung (C34.2), and one case of throat (C32.8) cancers. Neoplasms caused the death of four ARS survivors and nine patients with ARS-NC. For both groups the time from the initial diagnosis of cancer and the onset of death was 1.3 ± 1.3 years, the age at a time of death 63.6 ± 13.2 years.

Leukemia and oncohematological disorders were diagnosed in eight survivors (six ARS and two ARS-NC). During the postaccidental period amongst ARS1-3 survivors, three cases of myelodysplastic syndrome (D46.1, D46.4, D46.9), one case of acute myelomonoblastic leukemia (C92.5), one case of chronic myeloid leukemia (C92.1), and one case of non-Hodgkin B-cell lymphoma (C83.8) were revealed. All patients died, but in case of non-Hodgkin lymphoma myocardial infarction was the reason of lethal outcome. In two ARS-NC patients hypoplasia of hematopoiesis (C96.9) and polycythemia vera (D45) developed that brought them to death. The patient with osteomyelofibrosis that transformed into acute myeloid leukemia (C92.0) is still alive. The difference between numbers of blood disorders in ARS survivors group (6.6%), and ARS NC (2.0%) was insignificant (*p* > 0.05).

The mean survival period from diagnosis to death was 2.0 ± 2.1 years in both ARS survivors and ARS-NC patients. The mean age of death from oncohematological disorders was 57.9 ± 6.9 years. Analysis showed that 45% of ARS1-3 and ARS-NC patients died at the age of less than 62.3 years, which, according to the WHO data, was the average life expectancy of males in Ukraine.

## 4. Discussion

To quantify the risks of radiogenic cancers caused by Chornobyl exposure and to find out whether it differs from the estimates received for the Japanese A-bomb survivors, the following analytical studies were initiated: A case-control study on leukemia and a case-control study on thyroid cancer in cleanup workers in Ukraine. To reveal missing leukemia cases, a special registry was created containing 41,000 patients of the same age, gender, and inhabitance areas with 99 hematological disorders that might resemble leukemia. All the diagnoses were re-evaluated by the national review group and sent for international pathology expertise.

It has been widely recognized since early 1950′s Japanese studies that ionizing radiation may induce most types of leukemia, excluding chronic lymphocytic leukemia (CLL) [1,10]. A case-control study of Chornobyl cleanup workers in Belarus, Russia, and the Baltic countries demonstrated an elevated radiation-associated risk for CLL, though not statistically significant [18].

Moreover, the most recent atomic bomb survivor leukemia incidence study, showed a statistically significant linear dose-response for CLL [4]. Dose-related association between exposure to ionizing radiation and the increased incidence of CLL was also demonstrated in studies of occupationally exposed workers [19,20,21].

In Ukraine, a nested case-control study in a cohort of 110,645 male cleanup workers was performed together by the NRCRM and U.S. National Cancer Institute. Case identification and validation was performed by the international pathology review. Doses for cases and controls were reconstructed by a new RADRUE method [22,23]. For first 15 years, ERR/Gy value was 3.44 (95% CI 0.47; 9.78; *p* < 0.01) with surprisingly equal value for chronic lymphocytic leukemia and leukemia excluding CLL [24]. Further analysis for a 20 year period was based on an extended number of cases (160, among them 89 CLL). A significant linear dose response for 137 leukemia cases with reconstructed doses was identified, ERR/Gy = 1.26 (95% CI: 0.03–3.58). A detailed analysis has shown difficulties in a questionnaire-based dose reconstruction by RADRUE in 20 participants interviewed in a period less than 2 years after chemotherapy (ERR/Gy = −0.47 (95% CI: <−0.47 to 1.02)), presumably due to cognitive problems. For the remaining 117 cases, the ERR value was 2.38 (95% CI: 0.49–5.87), and a clearly demonstrated dose dependent excessive risk was demonstrated for either leukemia as a whole or leukemia subtypes separately. For CLL, the ERR/Gy was 2.58 (95% CI: 0.02–8.43), and for non-CLL, ERR/Gy was 2.21 (95% CI: 0.05–7.61). In total, 16% of leukemia cases (18% of CLL, 15% of non-CLL) were attributed to radiation exposure [25]. In a study of factors other than radiation after adjusting for radiation exposure, we identified a two-fold elevated risk for non-CLL leukemia for occupational exposure to petroleum, OR = 2.28 (95% CI: 1.13–6.79), mostly due to particularly high risk for myeloid leukemia. No associations with risk factors other than radiation were found for chronic lymphocytic leukemia [26].

In addition to CLL risk estimates, it was defined that older age at first exposure, smoking, and higher frequency of visits to the doctor were significantly associated with a shorter latent period. At the same time, the association of radiation dose and younger age at first radiation exposure at Chornobyl with shorter survival after diagnosis was shown, though not statistically significant [27].

Our present data on leukemia and lymphoma provide support to the mentioned studies as well as INWORKS analysis in radiation workers [9], and extend the elevated risks period up to 30 years after exposure. These results are in line with Japanese hibakusha data [1,2].

Thyroid cancer, along with leukemia, is the earliest manifestation of radiation exposure. The follow-up on the thyroid cancer frequency in a cohort of 150,813 male Chornobyl cleanup workers was launched in 1986 and originally continued through 2010 There were 196 followed-up incident thyroid cancer cases in the study cohort, with an overall SIR of 3.50 (95% CI: 3.04–4.03). A significantly elevated SIR estimate of 3.86 (95% CI: 3.26–4.57) was calculated for the cleanup workers who had their first cleanup mission in the Chornobyl zone in 1986 [28].

Continuation of the follow-up through 2012 [did not substantially change the values (Table 2). There were 216 incident thyroid cancer cases in the study cohort with an overall SIR of 3.35 (95% CI: 2.91–3.80). Elevated thyroid cancer incidence was detected in male cleanup workers who participated in cleanup activities during the entire period of cleanups (1986–1990), although it was statistically significant only among those who participated the activities held in 1986–1987, most possibly due to the difference in the dose (Table 2) [29].

In order to preliminarily estimate the dose-dependent risk of thyroid cancer in the cohort, and to calculate possible contribution of the radiation factor into the incidence rate, the doses to thyroid were assessed in two alternative ways using different sources of raw dosimetric data.

According to the first approach, the official dose records (ODR), which are available in SRU and were published in the UNSCEAR 2008 report [10] were used as a starting point for dose estimation. These whole-body doses were adjusted for known bias [30] and converted to doses for thyroid using the relevant conversion coefficients established by International commission on Radiological Protection (ICRP) [31].

Alternative thyroid dose assessment was based on the average RADRUE red bone marrow doses estimated in cleanup workers [16,17]. Red bone marrow doses available for 1000 study subjects were converted to thyroid doses, also utilizing aforesaid ICRP conversion coefficients. In both approaches the air kerma was used as intermediate reference value (i.e., whole body or red bone marrow doses were first converted to air kerma and then to thyroid doses). Excessive absolute risk—per 10,000 person-years, Gray (EAR/10^4^Gy), according to alternative dose assessment options, was estimated to be in a range from 1.86 (95% CI: 0.47–3.24) to 2.07 (95% CI: 0.53–3.62). The excess relative risk per Gray (ERR/Gy) ranged from 2.38 (95% CI: 0.60–4.15) to 2.66 (95% CI: 0.68–4.64) [29].

These estimates confirm the presence of a dependency between the radiation dose e and thyroid cancer in those exposed in adulthood. A more sophisticated NRCRM–NCI thyroid cancer case-control study nested in the cohort of 150,813 male Chornobyl cleanup workers is at the final stage of accomplishment. The field work including subject tracing and interviewing, dose reconstruction, as well as statistical risk analysis has been finalized. New estimates of the thyroid cancer risk in cleanup workers will be obtained soon.

In general, the NRCRM and NRCRM–NCI studies demonstrate radiation risks of thyroid cancer in cleanup workers and are in line with both A-bomb survivors and the studies coordinated by the International Agency for Research on Cancer (IARC) (Table 3).

The last UNSCEAR analysis has shown that the total number of cases of thyroid cancer registered in the period 1991−2015 in males and females, who were under 18 in 1986 (for the whole of Belarus and Ukraine, and for the four most contaminated oblasts of the Russian Federation), approached 20,000, and basically increased monotonically over the period of 2006−2015. The observed increase in the incidence of thyroid cancer is attributable to a variety of factors: Increased spontaneous incidence rate with aging of the birth cohort, radiation exposure, awareness of thyroid cancer risk after the accident, and improvement of diagnostic methods to detect thyroid cancer. The committee estimated that the fraction of the incidence of thyroid cancer attributable to radiation exposure is of the order of 0.25, with the uncertainty range of the attributable fraction extending at least from 0.07 to 0.5. In the opinion of the committee, the increased incidence of thyroid cancer after the Chornobyl accident is a major issue and needs further investigation to determine the long-term consequences of radiation exposure [37].

The studied postaccidental period demonstrated an increase in rates of “early” cancer types—thyroid, breast, and leukemia. Based on the experience of A-bomb survivors, cohort cancer rates in the studied cohorts will remain elevated due to aging and differences in radiation risks for specific cancer types. Hence, malignant disease monitoring is still an actual task of medical surveillance for the exposed cohorts.

### Pathology of Cancers in Radiation Exposed Population after Chornobyl

The first results from the radiation-induced cancers pathology were obtained in thyroid cancer, urinary bladder cancer, breast cancer, and leukemia. Attempts were done to find radiation specific cancer markers, but to date these have not been successful. Similar types of point mutations were described in different cancers, supposing the contributive role of radiation exposure (*BRAF*, *MAPK38*, *RAS*) [38,39].

Studies of leukemia have not demonstrated specific pathologic features of radiation-induced myelogenous and acute leukemia. As epidemiologic studies of cleanup workers have shown, regarding radiation risks of chronic lymphocytic leukemia, there was an interest in finding the specific features of radiation-induced CLL. A special substudy was performed on clinical and morphological features of CLL in the cleanup workers (80) in comparison with nonexposed CLL cases (70). The shorter period of white blood cell doubling in peripheral blood (10.7 vs. 18.0; *p* < 0.001), frequent infectious episodes, lymphadenopathy and hepatosplenomegaly (37 vs. 16), higher expression for CD38, and lower for ZAP-70 antigens were among the peculiarities, although not statistically significant.

Higher frequencies (89.3%) of unmutated immunoglobulin variable heavy chain (IGHV) genes were shown among 28 cleanup workers from 1986 with CLL, in comparison with 68.1% in 238 nonexposed CLL patients. In a later study, the same group of investigators failed to demonstrate any difference [40]. Comparison of genome changes in general exposed population vs. nonexposed one in the post-Chornobyl period describes upregulation of *MYC*, *HNF1A*, and *HNF4A* and *YWHAG*, *NF-κB1,* and *SP1*, together with a downregulation of *CEBPA*, *YWHAG*, *SATB1*, and *RB1* [41].

A further CLL cleanup workers study of 17 CLL cases from the cohort showed more frequent mutations in the telomere maintenance pathway genes POT1 and ATM, compared with 28 unexposed CLL patients from Ukraine and 100 from the USA. Tumor telomere length was significantly longer in cleanup workers and was associated with the *POT1* mutation and survival [42].

The performed molecular studies suggest the presence of a changed gene profile after exposure that could form a background for later effects, especially cancers. A statistically significant and dose-dependent decrease in expression of the *BCL2*, *SERPINB9*, *CDKN2A*, and *STAT3* genes was demonstrated in parallel to a dose-dependent overexpression of *MCF2L* and upregulation of *TP53* (up to 100 mSv). *IL1B* expression was the highest in exposed doses from 0.1 to 100 mSv with a negative correlation between at IL1B expression and CD19+3−, CD3-HLA-DR+, CD4+8− cell counts, and CD4+/CD8+ ratio. Hyper expression of *TNF* gene in doses above 100 mSv to 1,000 mSv was shown, and in higher doses a combination of *TNF* downregulation with an increase in *IFNG* gene expression were demonstrated with correlations with numbers of CD3+16+56+ and CD25+ lymphocytes, and inhibition of expression of *CLSTN2*. An increased expression of γ-H2AX and Cyclin D1 correlated to radiation dose, telomere shortening to age, and concomitant pathology.

## 5. Conclusions

Thirty years after the Chornobyl accident an excess was demonstrated in incidences of the “early” cancers—thyroid, breast, and leukemia, with a slight tendency to decrease at a later period. Dose dependency was shown for thyroid cancer and leukemia, and surprisingly radiation risks of chronic lymphocytic leukemia. For breast cancer incidence there are indications of an increase, but none of analytical case-control studies are available in cleanup workers and the general exposed population. Summarizing the incidence rates and risks, it should be stated that observed tendencies, dynamics, and risks magnitude are different for the malignancies of different localizations and consistent with those for other exposed populations.

The performed studies demonstrate the possibility of understanding the nature of radiation-induced effects after low-dose exposure. Genome instability, including elevated micronuclei counts, gamma-H2AX expression, telomere length variability, and changes in gene expression, could serve as background for low-dose health effects. To connect possible late effects, such as specific subtypes of radiogenic cancers, with radiation exposure, the analytical cohort and case-control studies need to include biomarkers of dose and disease supplemented by a uniform dosimetry.

Since the solid cancer risks in the studied cohorts supposedly have not been realized completely, the monitoring and registers for support are still an actual task of medical follow-up.

## Figures and Tables

**Table 1 cancers-11-01291-t001:** Cancer standardized incidence ratios, SIRs, (95% CI), in Ukrainian Chornobyl cleanup workers (1986–1987, both genders) by follow-up period and cancer site.

Cancer Site	Period of Follow-Up				
	ICD–10	1994–1999	2000–2005	2006–2010	2011–2016
All cancers	C00–C96	138.3	107.1	103.3	102.6
		(132.5–144.0)	(103.7–110.4)	(99.9–106.7)	(99.2–106.0)
Leukemia and lymphoma	C81–C96	232.6	201.8	123.9	140.8
		(200.9–264.3)	(180.0–223.7)	(105.4–142.4)	(121.2–160.6)
Thyroid cancer	C73	554.9	666.7	322.2	250.3
		(440.9–668.9)	(569.8–763.5)	(250.2–394.1)	(192.9–307.8)
Breast cancer	C50	185.2	176.1	140.3	130.4
		(143.3–227.1)	(146.9–205.3)	(113.0–167.7)	(103.0–157.8)

**Table 2 cancers-11-01291-t002:** Number of thyroid cancer cases, person-years of observation, and SIR in the cohort of male Ukrainian cleanup workers (150,813) by year of first mission in the Chornobyl zone [29].

Year of First Mission	Number of Cleanup Workers	Person-Years	Thyroid Cancer Cases		SIR (95% CI)
			Observed	Expected	
1986	93,819	1,337,478	148	40.5	3.65 (3.07–4.24)
1987	24,818	393,025	31	11.1	2.79 (1.81–3.78)
1988–1990	21,012	310,685.5	17	9.4	1.81 (0.95–2.67)
Subtotal 1986–1990	139,649	2,041,188.5	196	61.0	3.21 (2.76–3.66)
Unknown	11,813	95,220.5	20	3.4	5.88 (3.30–8.46)
Total	150,813	2,136,409	216	64.4	3.35 (2.91–3.80)

**Table 3 cancers-11-01291-t003:** Cancer risks in Ukrainian Chornobyl cleanup workers in comparison with other studies.

Study Group, Country, Reference	Type of Study, Cohort Size	Follow Up Time Period	Number of Cases(/Controls)	Risks
Leukemia (All types) C91–C95				
Chornobyl clean-up workers, Ukraine	Case-control			ERR Gy^−1^
[24]		1986–2000	71/501	3.44 (0.47–9.78; *p* < 0.01)
[25]	Case-control	1986–2006	137/863	
				ERR Gy^−1^
				2.38 (95% CI: 0.49–5.87; *p* < 0.04)
Leukemia (CLL excluded) C91.0, C92–C95				
Chornobyl cleanup workers, Russia [32]	Cohort	1986–1997	51	ERR Gy^−1^
	53,772			4.98 (95% CI: 0.59–14.47)
		1998–2007	60	
				−1.64 (95% CI: −2.55 to 0.57)
Life Span Study cohort, Japan [4]	Cohort	1950–2001	312	ERR Gy^−1^
	113,011			4.7 (95% CI: 3.3–6.5)
Chornobyl cleanup workers, Russia, Belarus, Baltic countries [18]	Case-control	1990–2000	19/83	ERR 0.1 Gy^−1^
				0.50 (90% CI: −0.38 to 5.7)
Chronic lymphocytic leukemia C91.1–C91.4				
Chornobyl cleanup workers, Ukraine [25]	Case-control	1986–2006	65	ERR Gy^−1^ 2.58 (95% CI: 0.02–8.43)
Chornobyl cleanup workers, Ukraine [33]	Cohort, 152,520	1987–2012	146	SIR 1.44
				(95% CI: 1.21–1.68)
Multiple myeloma (C90)				
Chornobyl cleanup workers, Ukraine [34]	Cohort, 152,520	1996–2013	69	SIR 1.38 (95% CI: 1.06–1.71)
A-bomb survivors [4]	Cohort study, 113,011	1950–2001	136, including 31 not exposed	ERR Gy^−1^ 0.38 (95% CI: −0.23 to 1.36), *p* = 0.21
All solid cancers (C00-C80)				
Ukrainian cleanup workers of 1986–1987 [35]	Descriptive	1994–2013	11,116	SIR 107.5 (95% CI: 105.4–109.6)
	84,599			
		1994–2014	11,666	
				SIR 106.9 (95% CI: 105.0–108.9)
A-bomb survivors [2]	Cohort	1958–2008	17,448	35% per Gy (90% CI 28%; 43%) increase for men; 58% per Gy (43%; 69%) increase for women
	105,427			
Female breast cancer (C50)				
Ukrainian cleanup workers of 1986–87 [35]	Descriptive	1994–2013	336	SIR 157.8 (95% CI: 141.0–174.7)
	11,300			
		1994–2014	351	SIR 156.7 (95% CI: 140.3–173.1)
A-bomb survivors [2]	Cohort	1958–1998	1073	ERR Gy^−1^ 0.87 (90% CI: 0.55–1.30)
	105,427			
Thyroid cancer (C73)				
A-bomb survivors [2]	Cohort	1958–1998	471	ERR Gy^−1^ = 0.57 (90% CI: 0.24–1.10)
	105,427			
Cleanup workers [36]	Case-control	Russia: 1993–1998; Belarus: 1993–2000; Baltic: 1990–2000	107/423	ERR per 100 mGy = 0.38 (95% CI: 0.10–1.09)
Ukrainian Cleanup workers [28]	Cohort: 150,813;	1986–2010	196	SIR = 3.50 (95% CI: 3.04–4.03)
	Cohort: 150,813;	1986–2012	216	ERR Gy^−1^
[29]				from 2.38 (95% CI: 0.60–4.15) to 2.66 (95% CI: 0.68–4.64)

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
