# Peer review of "Cancers after Chornobyl: From Epidemiology to Molecular Quantification"

_cancers, 2019, doi:10.3390/cancers11091291_

Round 1

Reviewer 1 Report

The paper represent an overview of epidemiology, pathology and molecular findings studying post-Chernobyl cancers. It is suggested to shorten the section on ARS survivors outcomes and give more attention on breast cancer findings where possible. Specific comments are given on the enclosed pdf file of the manuscript.   

Author Response

Authors are grateful for a review and agree with all proposed changes. Changes were done to a revised manuscript according to .pdf file sent by reviewer

Reviewer 2 Report

Interesting paper with relevant information about the risk of malignancies after the Chernobyl accident. The paper presents both history, epidemiology and molecular biology.

The references and information is ok, but the major problem with this paper, as I can see, is the obscure purpose and disposition. Is it a review paper or do the authors present new data/results also?

-The introduction is very/too short.

- The intro doesn't address a proper aim. Row 32 p 1 says that "it is a systematic review", but it isn't only a review.  The paper appears to be a mixture between review and new unpublished results from the research center, since there seem to be unpublished data from the research group along with reviewing previously published results. Choose review or a paper of results.

-section 2 "Study subjects and methods" is a review of previous studies. Why not call it review?

-section 3 is a mixture of "results and discussion". Why mix these topics up?

-It is unclear if data in table1, table 4 and table 5 refers to previous results or if the authors present new unpublished data in these tables and the text. If this paper presents new results it should be made clear.

-What are the numbers 1 to 7 below the heading in Table 5 ment for?

-What is the purpose of presenting a list of cancer cases in the ARS survivors/group? (Table 4 and 5) Are these diagnoses part of the cohort/or case-control studies previously published? Why not present risk estimates to these cases?

Author Response

Authors agree with revisions and included them in italics to the attached file and to a revised paper

Reviewer 3 Report

The numerical data are important and can be published. 

The problem about this manuscript is that it takes for granted that surplus cancers after the Chernobyl accident were radiation-induced. However, the UNSCEAR 2008 Report concluded that no cancer incidence increases, other than thyroid cancer in patients exposed during childhood or adolescence, can be attributed to irradiation from the Chernobyl accident [1]. 

The Chernobyl accident provides an example of considerable difference in diagnostic quality of many diseases before and after the accident. The registered incidence e.g. of pediatric thyroid cancer prior to the Chernobyl accident was low in comparison to other developed nations obviously due to the differences in the diagnostic quality and coverage of the population by medical checkups. Therefore, the screening in the contaminated territories after the Chernobyl accident was yielding. In addition to the screening-effect, improved registration and reporting, the following mechanisms probably contributed to the cancer incidence increase after the accident: registration as Chernobyl victims of patients brought from non-contaminated areas, classification of questionable and borderline lesions as cancers as well as some percentage of false-positivity [2,3]. 

One of the mechanisms of the false-positivity was as follows. If a thyroid nodule is found by the screening, a fine-needle aspiration (FNA) is usually performed. Aspiration cytology of the thyroid is associated with a relatively high percentage of uncertain conclusions (so-called grey zone), when histological verification is indicated. Hemithyroidectomy or subtotal thyroidectomy was usually performed in such cases, and the surgical specimen sent to a pathologist, who could be sometimes prone, after the in toto removal of the nodule, to confirm malignancy even in case of some uncertainty. The FNA was introduced into practice later than ultrasonography, which additionally contributed to the overdiagnosis during the 1990s [2]. 

As for the clean-up workers (liquidators), it is hard to believe that surveillance bias was not operating. Regular annual medical examinations were offered to the liquidators [1]. The counterpart of the clean-up workers in the population – middle-aged men – are generally not covered by medical examinations. The autopsy rates must have been higher in liquidators than in the general population, which resulted in additional cancers diagnosed post mortem. The attenuation of the excessive risk later on after the accident was probably caused by the subsiding post-Chernobyl “radiation phobia” and the screening activity.

In contrast to the data from Russia and Ukraine, the study of a liquidator cohort from Lithuania, Latvia and Estonia revealed “no increased proportional incidence ratios for leukemia or radiation-related cancer sites combined” [4], which probably reflected the better diagnostics and coverage of the general population by medical checkups in the above-named countries. The Baltic study revealed “an elevated proportion of thyroid cancers in relation to the general male population; however, this finding could be attributable, at least in part, to more intensive medical examination, including thyroid screening” [4]. In general, Baltic liquidators were found to be “at small to no increased cancer risk after 22 years of observation” in contrast with “claims of substantially higher radiation risks in other populations” [4].

An example (lines 198-204 of the manuscript):

“The radiation dependent lesions in the urinary bladders of people living in cesium 137 (137Cs) 198 radio contaminated areas of the Ukraine were studied in biopsies from 159 male and 5 female 199 patients. A pattern of chronic proliferative atypical cystitis accompanied with large areas of sclerosis 200 of connective tissue in the lamina propria was commonly observed in all cases. These lesions were 201 associated with a dramatic increase in the incidences of dysplasia/carcinoma in situ, and, moreover, 202 small urothelial carcinomas were incidentally detected… [5].” 

The high prevalence of the bladder dysplasia/carcinoma in situ in randomly selected patients with benign prostatic hyperplasia [5] was deemed unrealistic and indicative of the false-positivity that probably resulted in overtreatment and over-manipulation including cystoscopies with biopsies [2].

Considering the above, the subtitle “Pathology of radiation-induced cancers after Chornobyl” is potentially misleading as it is unproven that a majority of cancers under discussion were indeed radiation-induced. Histopathological and other special features of Chernobyl-related cancers could be partly explained by a misinterpretation of old neglected cancers, found by the screening after the accident, as aggressive radiation-induced tumors [2,3]. The misinterpretation caused excessively radical surgical treatments of thyroid cancers after the Chernobyl accident especially in Belarus [3]. This matter should be discussed; the subtitle better to change to “Pathology of Chernobyl-related cancers”.

The authors may express their own opinion but they should discuss other published interpretations of analogous data [2,3].

References

1. United Nations Scientific Committee on the Effects of Atomic Radiation (UNSCEAR) (2008) Sources and effects of ionizing radiation. Report to the General Assembly, volume 2. Annex D. Health effects due to radiation from the Chernobyl accident. United Nation, New York

2. Jargin SV. Chernobyl-Related Cancer and Precancerous Lesions: Incidence Increase vs. Late Diagnostics. Dose Response. 2014;12(3):404-14. 

3. Jargin SV. Chernobyl-related thyroid cancer. Eur J Epidemiol. 2018;33(4):429-431. https://rdcu.be/KTdK 

4. Rahu K, Hakulinen T, Smailyte G, et al. Site-specific cancer risk in the Baltic cohort of Chernobyl cleanup workers, 1986–2007. Eur J Cancer 2013;49(13):2926–2933.

5. Romanenko A, Morimura K, Wanibuchi K, et al. Urinary bladder lesions induced by persistent chronic low-dose ionizing 364 radiation. Cancer Sci 2003; 94 (4): 328–333.

Author Response

Answers to reviewer's comments are in italics ia an attached file. We have revised a  manuscript accordine to comments.

Round 2

Reviewer 3 Report

The article cn be published s it is.

Page 1.

“ Chronic lymphocytic leukemia risks were not analyzed as such pathology is absent in ethnic Japanese”

“Absent” should be replaced by “relatively rare” or alike [1].

1. Tamura K, Sawada H, Izumi Y, Fukuda T, Utsunomiya A, Ikeda S, Uike N, Tsukada J, Kawano F, Shibuya T, Gondo H, Okamura S, Suzumiya J; Kyushu Hematology Organization for Treatment (K-HOT) Study Group. Chronic lymphocytic leukemia (CLL) is rare, but the proportion of T-CLL is high in Japan. Eur J Haematol. 2001;67(3):152-7.

This manuscript is a resubmission of an earlier submission. The following is a list of the peer review reports and author responses from that submission.